# The Effectiveness of Different Concepts of Bracing in Adolescent Idiopathic Scoliosis (AIS): A Systematic Review and Meta-Analysis

**DOI:** 10.3390/jcm10102145

**Published:** 2021-05-15

**Authors:** Lorenzo Costa, Tom P. C. Schlosser, Hanad Jimale, Jelle F. Homans, Moyo C. Kruyt, René M. Castelein

**Affiliations:** Department of Orthopaedic Surgery, University Medical Center Utrecht, 3508 GA Utrecht, The Netherlands; L.Costa-2@umcutrecht.nl (L.C.); T.P.C.Schlosser@umcutrecht.nl (T.P.C.S.); h.jimale@students.uu.nl (H.J.); J.F.Homans-3@umcutrecht.nl (J.F.H.); M.C.Kruyt@umcutrecht.nl (M.C.K.)

**Keywords:** systematic review, meta-analysis, adolescent idiopathic scoliosis, brace therapy, brace concepts, rigid brace, night time brace

## Abstract

Brace treatment is the most common noninvasive treatment in adolescent idiopathic scoliosis (AIS); however it is currently not fully known whether there is a difference in effectiveness between brace types/concepts. All studies on brace treatment for AIS were searched for in PubMed and EMBASE up to January 2021. Articles that did not report on maturity of the study population were excluded. Critical appraisal was performed using the Methodological Index for Non-Randomized Studies tool (MINORS). Brace concepts were distinguished in prescribed wearing time and rigidity of the brace: full-time, part-time, and night-time, rigid braces and soft braces. In the meta-analysis, success was defined as ≤5° curve progression during follow-up. Of the 33 selected studies, 11 papers showed high risk of bias. The rigid full-time brace had on average a success rate of 73.2% (95% CI 61–86%), night-time of 78.7% (72–85%), soft braces of 62.4% (55–70%), observation only of 50% (44–56%). There was insufficient evidence on part-time wear for the meta-analysis. The majority of brace studies have significant risk of bias. No significant difference in outcome between the night-time or full-time concepts could be identified. Soft braces have a lower success rate compared to rigid braces. Bracing for scoliosis in Risser 0–2 and 0–3 stage of maturation appeared most effective.

## 1. Introduction

Idiopathic scoliosis is a deviation from normal growth of the spine and trunk, with a prevalence of 2–4% in the general population [1]. Its management depends on the magnitude of the spinal curvature. Observation is indicated for mild curves and brace treatment is normally recommended in curves between 20° and 45° [2,3]. The application of many different brace concepts (distinguished in prescribed wearing time and rigidity of the brace: full-time, part-time, and night-time, rigid braces and soft braces) have been described in the literature. They all apply different degrees of external corrective forces to the trunk to correct the complex 3-D spinal deformity. Full time braces usually aim at in-brace correction of the curve to at least 50% of the original magnitude; nighttime braces are a bit more ambitious and aim to correct about 70% or an even higher percentage of the curve while the brace is worn [4,5].

The “Bracing in Adolescent Idiopathic Scoliosis Trial” (BRAIST) has provided high-quality evidence for the application of full-time rigid brace treatment in AIS patients with curves 20–40° before skeletal maturity [6]. For other concepts of bracing, most studies are retrospective and not controlled [7]. Despite the efforts of societies like SRS and SOSORT, high-quality evidence for the effectiveness of other concepts of bracing is still lacking. Furthermore, due to the development of multiple braces and non-standardized criteria, it is difficult to compare the results. Nevertheless, many studies provide insight in effectiveness [6,8,9].

The aim of this systematic review and meta-analysis is to evaluate the literature on the effectiveness of different concepts of brace treatment, in terms of effect on spinal curve magnitude. The questions are:What is the most effective brace concept?What is the most effective brace type (Boston brace, Providence brace, etc.)?What is the effect of skeletal maturity on the effectiveness of different concepts of brace treatment?

Even though Randomized Control Trials (RCTs) are considered to be more reliable than observational studies when evaluating treatment effectiveness, RCTs are extremely demanding for these types of questions and often fail [10]. Meta-epidemiological research has shown that for non-pharmaceutical purposes, alternative study designs are not consistently more biased and should not be discarded. Therefore, we also included observational studies [11,12,13,14]. To allow assessment of this wide array of studies, tools are available to appraise study quality for non-comparative studies [15].

Lastly, many definitions of success rate such as ≤5°, ≤10°, or avoidance of surgery are used in scoliosis brace studies. As this heterogeneity would have affected the outcome of this review, the authors agreed to use, at least for the meta-analysis, the most used definition: ≤5° of curve progression as successful treatment.

## 2. Materials and Methods

### 2.1. Protocol

This systematic review was performed according to the PRISMA statement and is registered at PROSPERO with the ID CRD42020157636 [16].

### 2.2. Search Methods and Study Selections

A systematic search was undertaken to identify all studies reporting on bracing in AIS in PubMed and EMBASE till January 2021 (see Table 1). Inclusion criteria were shown in Table 2.

All studies that reported on spinal deformities other than AIS or with mixed-age population that did not report the outcomes for AIS separately were excluded. Since skeletal maturity is considered a significant parameter, if not stated, the studies were excluded from further analyses. Reviews, cross-sectional studies, and case series with less than 10 patients were also excluded (see Table 2). Title/abstract and full-text screening was done by two independent investigators. To ensure literature saturation, reference lists of included studies or relevant reviews identified through the search were reviewed.

### 2.3. Appraisal

Two authors independently assessed the quality and risk of bias of each included study using the validated Methodological Index for Non-Randomized Studies (MINORs) (see Table 3) [15]. For any disagreement, consensus was reached by discussion.

### 2.4. Synthesis

Brace concepts and brace types, prescribed wearing time, actual wearing time (if reported), rigidity of the brace, maturity parameters, age, sex, curve magnitude, and outcomes (effect on curve magnitude and prevention of the need of surgery) were systematically collected and compared between the different brace concepts/types. Since heterogeneity in quality, methodology, and outcomes of the different studies was expected, a best-evidence-synthesis was performed in the form of a systematic qualitative synthesis [18]. The qualitative synthesis describes the outcomes of non-comparative and comparative studies on the different brace concepts/types. Because the majority of the studies report on success rates defined as ≤5° progression during study follow-up, success rates are reported according to this definition (otherwise indicated in the text). Follow-up is intended after at least 1 year of follow-up and/or after termination of brace treatment.

The meta-analysis was performed on the outcomes of studies with low risk-of-bias that reported on the effectiveness of different brace concepts or braces as defined as ≤5° coronal curve angle progression during study follow-up. In addition, the same was made with ≤50° of Cobb angle progression. OpenMeta Analyst was used to execute the analysis [19]. Mean success rate and 95% confidence intervals (95% CI) were calculated and compared between the concepts. Only the BRAIST study included data on untreated patients. Due to lack of other studies with control groups, two studies not included in this review on the natural history of AIS were used for calculation of the success rate of observation only (*n* = 267) [20,21].

The effect of skeletal maturity on the success of the brace treatment was analyzed using the same criteria as the meta-analysis. Outcomes were compared between studies with different skeletal maturity at inclusion: Risser sign 0–1, 0–2, 0–3, and 0–4.

## 3. Results

### 3.1. Search

The search yielded a total of 2609 papers. The PRISMA flowchart and reason for exclusion are shown in
Figure 1. After title/abstract screening, 224 articles were selected for full-text reading. Reference tracking yielded no additional articles. After exclusions, a total of 33 articles were included in this study.

### 3.2. Study Characteristic

In total, seven types of rigid, full-time braces; two types of rigid, part-time braces; two types of rigid, night-time braces; and one soft, full-time brace were described in the 33 studies (see Table 4).

The study population varied between 23 and 843 patients per study. To one of the RCTs, patient preference cohorts were added during the inclusion period [6]. Seven studies (21%) recruited patients in Asia, seven (21%) in North-America and nineteen (58%) in Europe. All studies reported on prescribed wearing time and three studies reported on actual wearing time as assessed by a thermomonitor or by the orthoptist [6,42,43]. Concerning the Cobb angle, the inclusion criteria of the non-comparative studies were:Rigid full-time braces (15 studies): 30% 20–40°, 30% 25–40°, 8% 25–45°, 8% ≥ 40°, 8% > 25°, 8% 0–45°.Night-time braces (8 studies): 63% 25–40°, 13% 20–45°, 13% 25–49°, 13% < 25°.Soft full-time braces (2 studies): 50% 25–40°, 50% 15–40°.

For the comparative studies (8 studies): 50% 25–40°, 26% > 25°, 13% 20–30°, 13% 15–30° for soft braces and >30° for rigid full-time braces.

At the inclusion, most of the studies reported a magnitude between 25–40° for the Cobb angle. Two papers presented a mean Cobb angle above 40° and one paper below 20°.

In total, 69% of the studies included thoracic, thoracolumbar, and lumbar curves; 25% included thoracolumbar curves; and 6% double major curves.

The radiographic skeletal maturity at initiation of the treatment was reported in all papers. Of the studies, 32 used the Risser sign and one study only used menarche as a proxy for maturity [39]. Of those, 3% used the Risser sign between 0–1 (as subgroup), 70% between 0–2, 18% between 0–3, 3% between 0–4, 3% used Risser 0–1 and Tanner between 2–3, and 3% used Risser 0–2 and menarche period as well (pre-menarche or 1 year post-menarche) (see Table 5 and Table 6).

### 3.3. Study Quality

The critical appraisal results are shown in Table 5 and Table 6. The mean quality score of the 3 RCTs was 14.5 (out of 24); of the 8 comparative cohort studies 10.7 (out of 24); and of the 24 non-comparative cohort studies, 8.1 (out of 16). Twelve studies had low quality and high risk-of-bias and six did not report on success defined as ≤5° progression. A total of sixteen papers were included in the meta-analysis. There was insufficient evidence available to include the part-time, rigid brace concept in the meta-analysis. Quality of the two studies added for the control group was 7/16 and 15/24 [20,21].

### 3.4. Qualitative Analysis

Full-time, rigid braces

Thirteen studies described the outcomes of six full-time, rigid braces. Six brace types were individually studied (see Table 4).

Weinstein et al. (*n* = 242) included rigid full-time TLSOs (thoraco-lumbo-sacral orthosis) in a study that was designed as RCT to which patient preference cohorts were added (68% was treated with Boston brace) [6]. This was one of the few studies that defined success as progression to no more than 50° Cobb angle and no surgical treatment. Brace treatment was successful in 72% of cases versus 42% in the observation only group [6].

The effectiveness of the Boston brace was investigated in three studies [22,23,24]. The success rate was 51–83% in a total of 169 patients [22,24]. Yrjönen et al. found that 63% of boys had a successful treatment compared to 78% of the girls [23]. The Chêneau brace was investigated in four studies: Pasquini et al. (low quality study) reported a success rate of 81%, Fang et al. of 81% (defined as no curve progression to >50°) and Pham et al. of 86% (defined as curve progression ≤10°) [25,26,27]. Pham et al. indicated that the Chêneau brace was most effective in the lumbar curves [27]. Zabrowska-Sapeta et al. (*n* = 79), studied the Cheneau brace in combination with physiotherapy. Treatment was successful in 48% of cases [28]. Maruyama et al. (*n* = 33) investigated the Rigo-Chêneau brace. Success was observed in 76% of the patients [44]. The progressive action short brace(PASB) was studied by Aulisa et al. in 69 and 163 patients. In this study with low quality (6/16), the reported success rate was between 65.6 and 100% [29,45]. Similarly, the Lyon brace was studied by Aulisa et al. in 69 patients and the reported success rate was 99% [30]. The Gensingen brace was studied by Weiss et al. (*n* = 25). The percentage of successful treatment was 92% [31]. 

The Osaka Medical College Brace was studied by Kuroki et al. (*n* = 31). Treatment was successful in 68% of the patients [32].

Pressure-adjustable orthosis was developed by Yangmin Lin et al. in 2020 (*n* = 24). The reported treatment success was 100% after 1 year of treatment [33].

Part-time, rigid braces

There were no non-comparative studies.

Night-time, rigid braces

Eight studies focused on rigid, night-time braces with a total of 762 patients treated with the Charleston or Providence brace. Lauteur et al. (*n* = 142) studied the night-time brace concept. The treatment was successful in 83% of cases [46].

The Charleston brace was studied by Lee et al. (*n* = 95) and Price et al. (*n* = 139) with a success rate of 83% and 84% [34,35]. Price et al. noticed that patients with double curves treated with Charleston brace should be observed closely for the risk of increase in compensatory curves [34].

The Providence brace was studied in four non-comparative studies with low to moderate quality (total *n* = 56 + 63 + 102 + 34 + 80) [5,36,37,38,39]. Success rates were between 52% and 89% [36,39].

Full-time soft braces

Two studies by Coillard et al. reported on one type of full-time soft braces (SpineCor), one RCT (*n* = 68) compared to controls (study quality = 16/24) and one non-comparative cohort (*n* = 101).

With 5 years follow-up or follow-up to more than 2 years after discontinuation, the brace treatment was successful in 59–73% of the patients [40,41].

Comparative studies

Minsk et al. (*n* = 108) compared the Rigo-Chêneau brace and a thoraco-lumbo-sacral orthosis (TLSO). Success rate was defined as curve ≤5° and failure defined as need of surgery [47]. No patients with the Rigo-Chêneau needed surgery compared to 34% of patients with the TLSO brace that needed surgical intervention [47].

Two studies compared full-time with part-time braces (part-time was considered: ≤16 h—Hanks et al., as prescribed wearing time—or 7–12 h—Katz et al., as actual wearing time) [42,43]. Wilmington Jacket (Hanks et al. *n* = 100, success rate defined as curve progression ≤10°) and Boston brace (Katz et al. *n* = 100) were identified. The Wilmington Jacket full-time group had a success of 80%, while the part-time group had a successful treatment in 84% of the cases [42]. Boston full-time bracing wear had a success of 82% and the part-time worn brace of 31% [43].

Three studies, Yrjönen et al. (*n* = 72), Janicki et al. (*n* = 83), and Ohrt-Nissen et al. (*n* = 77), prospectively compared the Providence brace and the full-time TLSO [4,48,49]. Success rate was defined as curve progression ≤5° or residual curve < 45° [49]. Providence brace was successful in 31%, when the full-time TLSO was in 15% [4,48].

Two studies, Weiss et al. (*n* = 22) and Guo et al. (RCT *n* = 38), focused on the SpineCor brace vs TLSO [50,51]. Success was between 8 and 65% for the SpineCor and between 80% and the 94.4% for the TLSO [50,51].

### 3.5. Meta-Analysis

Sixteen studies had medium or low risk-of-bias, with defined success as progression ≤ 5° and were included in the meta-analysis. The rigid full-time brace had a success of 73.2% (95% CI 60.9–85.5%), the night-time of 78.7% (95% CI 72.4–85%), and soft braces of 62.4% (95% CI 55.1–69.6%) (see Figure 2). The success rate of observation was only 50% (95%, CI 44–56%) [20,21,41]. In addition, three studies over rigid full-time bracing with medium or low risk-of-bias, when success is defined as progression ≤ 50°, were separately included in the analysis. The success was of 73.2% (95% CI, 67.2–79.2%). The bubble plot shows no relation between the study quality and reported success rates (Figure 3).

### 3.6. The Role of Skeletal Maturity

Fourteen studies were included for the assessment of efficacy in relation to the Risser sign at initiation of brace therapy. Xu et al. divided their study group already based on Risser stage at initiation, so these subgroups were used separately for each category. For the category Risser 0–1, one paper could be used and the success rate was low, 42% (95%CI not applicable because there was only one study). For patients included with Risser 0–2, 10 papers reported a success rate of 71% (68–74%). The 0–3 category yielded 3 papers with a success rate of 75% (70–80%). The category that included all Risser stages at initiation of therapy showed a little lower success rate of 60% (54–66%) (see Figure 4).

## 4. Discussion

This systematic review and meta-analysis aimed to compare the effectiveness of different concepts of bracing for treatment of AIS. Comparison of the effectiveness between the different brace concepts revealed that rigid braces have better outcomes than soft braces and that night-time braces have comparable effectiveness compared to full-time braces. Rigid part-time bracing data were too limited to be included in the meta-analysis [4,42,48,49,50,51]. Most of the studies used as curve type thoracic, thoraco-lumbar, and lumbar curves with no subgroups organization. Therefore, it was not possible to detect differences in brace efficacy between different curve types. Nevertheless, most of the papers reported curves between 25–40° Cobb angle (no subgroups) as well as the Risser sign between 0–2 at initiation of the study and differences between brace types could be analyzed as seen in the results section.

A clear distinction can be made between rigid braces with a constant shape that force the body in a certain position and soft or dynamic orthoses that exert a constant force [52,53]. Another distinction is the prescribed wearing time. Full-time is the most used concept, where the patient should wear the brace 18 to 23 h per day. Part-time bracing is normally prescribed for less than 16 h per day. Night-time bracing is a different concept that aims to provide better correction because of the reduction of axial loading on the supine spine. [4,39].

Thirty-four studies could be selected for the purpose of this systematic review. Important reasons to exclude studies were: unclear or not stated range of coronal Cobb angle, etiology, assessment of maturity at inclusion, type of brace, follow-up time, time in brace, and the success/failure rate. The vast majority of the studies had a moderate to high risk of bias, as also shown by Negrini et al. in their review and SOSORT guidelines [8,18]. First, due to the nature of the treatment it is not possible to conduct trials in which patients are blinded. Second, it is difficult to perform RCTs in this kind of treatment because the risk of selection bias [54,55,56]. In 2008, Bunge et al. tried to perform an RCT study, however with no success since “it is harder to perform a RCT that abolishes or postpones a treatment than a RCT that adds a new treatment” [57]. In 2013, the BRAIST study was published (started as an RCT and ended as an RCT combined with patient preference cohorts). The patient’s/parent’s preference had led to a substantial proportion of patients refusing randomization and therefore decreasing the external validity. Nevertheless, the efficacy of bracing is generally accepted now because of this trial [6]. In the present study, we included 31 cohort studies and 3 RCTs. In general, estimates of treatment effectiveness are predominantly affected by the quality measures of the study design. In RCTs, for example, if randomization is not adequate, the effects of the treatment are overestimated [58,59]. Interestingly, methodological research has indicated that especially for non-pharmaceutical treatments the validity of observational studies is not necessarily inferior [60,61].

In line with the aim of bracing, success means the avoidance of surgery, as was used in the BRAIST study. However, since criteria for surgery vary between institutes and countries that definition of success is difficult for comparison. The most often used proxy for effectiveness in the available literature is the prevention of progression (≤5°). Despite its shortcomings, this outcome can be used for comparisons of different brace concepts and brace types. Ideally, future brace studies should report on (1) the percentage of patients who have ≤5° curve progression per year, at skeletal maturity and two years after ending brace, and the percentage of patients who have >5° progression up to skeletal maturity, (2) the percentage of patients with coronal curve angle exceeding 45° at skeletal maturity, (3) the percentage who have had surgery recommended/undertaken and (4) skeletal maturity parameters [62,63].

Skeletal maturity analysis show that the Risser grade, particularly stages 0–2, is still the most used classification for skeletal maturity assessment. This should be considered as the parameter for more homogeneous inclusion criteria in future studies. Moreover, our results highlight the correlation between maturity and chance of curve progression.

Furthermore, it is interesting to notice that if rigid full-time braces with success defined as no more than 5° progression is compared with the data of the same brace concept with definition of success as no more than 50° of ultimate Cobb angle, the results are comparable. This should need further analyses to understand its relevance since it is hard to objectively compare them.

In our opinion, this systematic review and meta-analysis provides a valuable addition to the existing literature.

To avoid heterogeneity of the data, future studies should also perform stratifications of the subjects related to initial Cobb angle, type of curve, sex, and skeletal maturity.

## 5. Conclusions

Bracing is effective in AIS treatment. Rigid full-time braces, rigid night-time braces, and full-time soft braces are more effective than observation only in terms of halting curve progression. The reported effectiveness of night-time braces is comparable to full-time rigid braces; soft braces perform less well. The Risser sign is still the most used parameter for bone maturity.

## Figures and Tables

**Figure 1 jcm-10-02145-f001:**
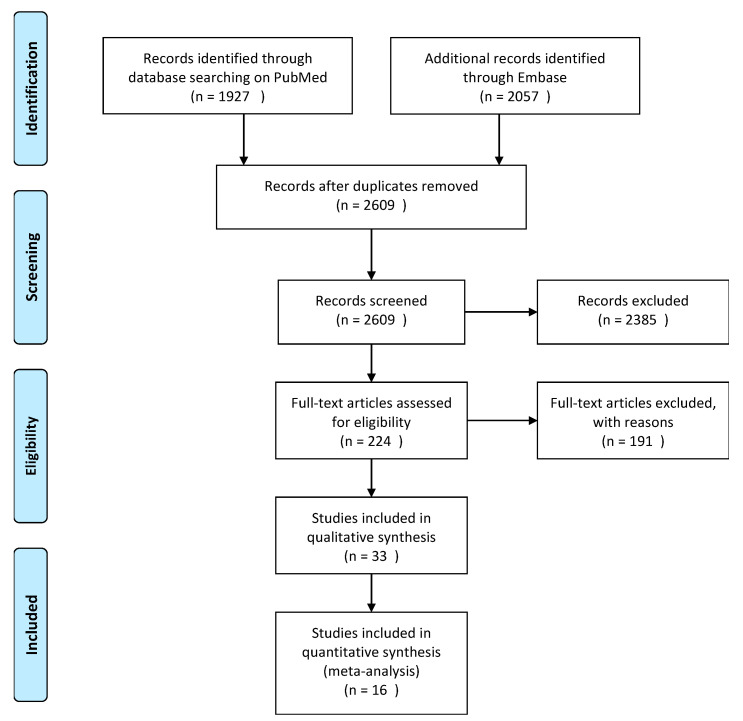
Flowchart of literature search.

**Figure 2 jcm-10-02145-f002:**
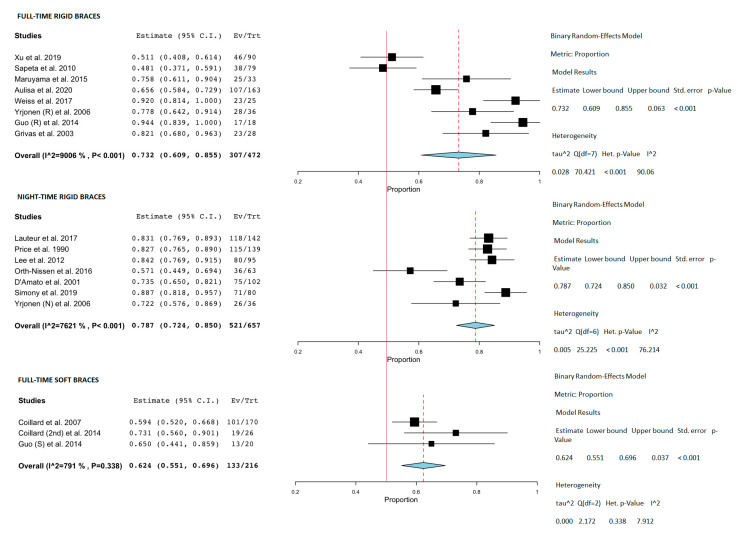
Forest plot of the studies divided per type of concept. Control groups were selected based on type of scoliosis (AIS). A success rate (≤5°) and reviews or case reports (<10 pt.) were excluded. The red line represents successes in the case control group (50%) [4,5,22,24,28,31,34,35,37,39,40,41,44,45,46,51].

**Figure 3 jcm-10-02145-f003:**
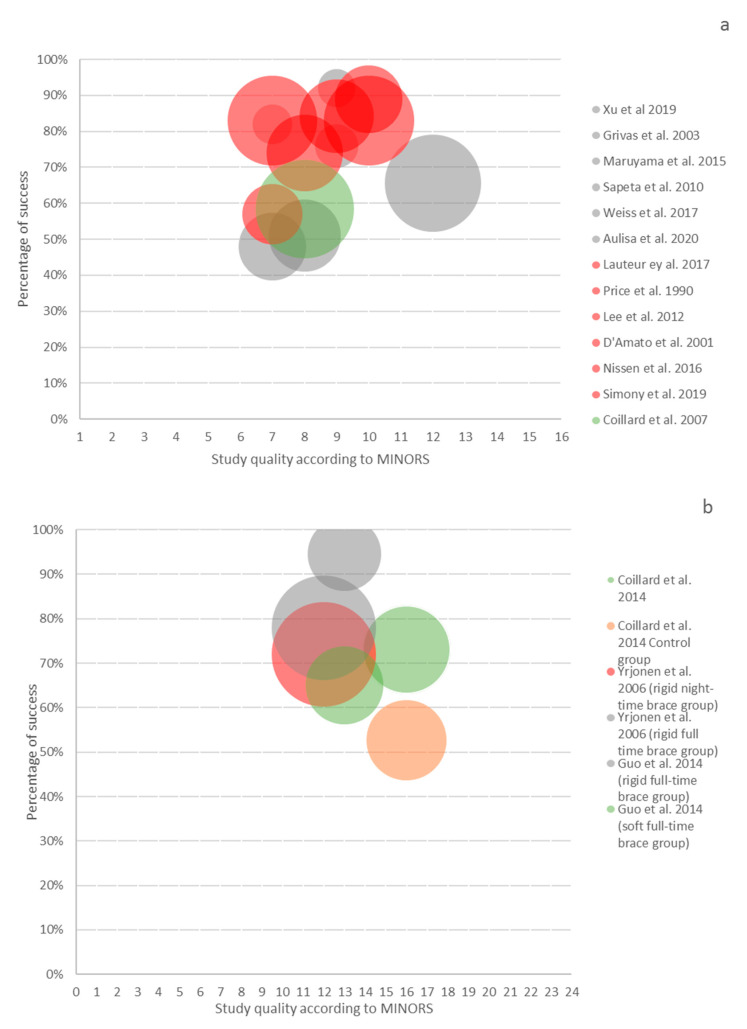
The bubble graph represents the proportion of the population with less than 5° Cobb angle progression (Y axis) relative to the MINORS score (X axis) and sample size (diameter of the circle). Thirteen non-comparative studies are represented in (**a**) and three comparative studies in (**b**). The grey color represents the rigid full-time concept, the red color the night-time concept, the green one the soft concept, and the orange color represents the control group [4,5,22,24,28,31,34,35,37,39,40,41,44,45,46,51].

**Figure 4 jcm-10-02145-f004:**
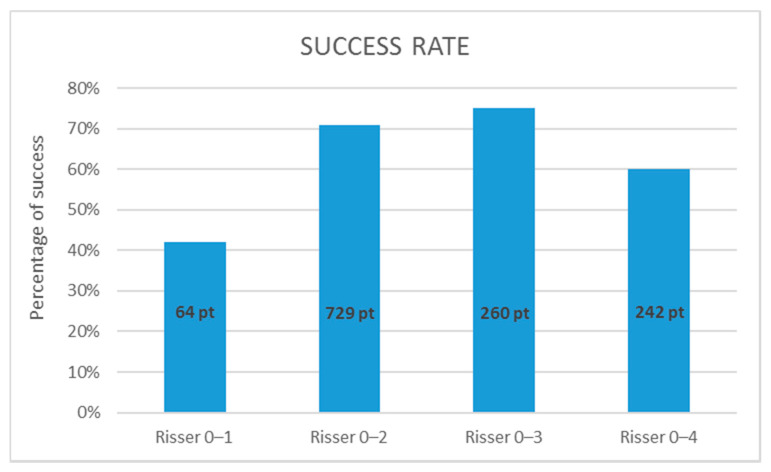
Mean success rate related to maturity. All studies used the Risser sign for assessment of skeletal maturity for initiation of brace treatment and used ≤5° as definition of successful treatment.

**Table 1 jcm-10-02145-t001:** Strategy of the search in PubMed and Embase. There was no language restriction. Duplicates were removed in Rayyan [17].

PubMed	(((scoliosis [MeSH Terms] OR scolio * [Title/Abstract] OR spinal curvature [Title/Abstract] OR AIS [Title/Abstract]))) AND ((((brace [MeSH Terms] OR brace [Title/Abstract] OR bracing [Title/Abstract]))) AND ((time [Title/Abstract] OR parttime [Title/Abstract] OR nighttime [Title/Abstract] OR compliance [MeSH Terms] OR compliance [Title/Abstract] OR compliant [Title/Abstract] OR effect [Title/Abstract] OR treatment * [Title/Abstract] OR result [Title/Abstract] OR results [Title/Abstract] OR therap [Title/Abstract] OR mental disorder [Title/Abstract] OR hypersensitive [Title/Abstract] OR peer problem [Title/Abstract] OR depress [Title/Abstract]) OR psychologic [Title/Abstract] OR quality of life [Title/Abstract] OR quality of life [MeSH] OR life quality [Title/Abstract])))).
Medscape	(‘scoliosis’: exp OR ‘scolio *’: ti, ab, kw OR ‘spinal curvature *’: ti, ab, kw OR ‘AIS’: ti, ab, kw) AND (‘brace’: exp OR ‘brace *’: ti, ab, kw OR ‘braci *’: ti, ab, kw) AND (‘time’: ti, ab, kw OR ‘parttime’: ti, ab, kw OR ‘nighttime’: ti, ab, kw OR ‘compliance’: exp OR ‘compliance’: ti, ab, kw OR ‘compliant’: ti, ab, kw OR ‘effect *’: ti, ab, kw OR ‘treatment *’: ti, ab, kw OR ‘result’: ti, ab, kw OR ‘results’: ti, ab, kw OR ‘therap *’: ti, ab, kw OR ‘mental disorder *’: ti, ab, kw OR ‘hypersensitiv *’: ti, ab, kw OR ‘peer problem *’: ti, ab, kw OR ‘depress *’: ti, ab, kw OR ‘psychologic *’: ti, ab, kw OR ‘quality of life’: ti, ab, kw OR ‘quality of life’: exp OR ‘life quality’: ti, ab, kw)

**Table 2 jcm-10-02145-t002:** Details regarding inclusion criteria.

**1**	Design	Longitudinal studies with at least one-year follow-up from brace initiation
**2**	Population	Patients with adolescent idiopathic scoliosis
**3**	Intervention	Specification of the concept(s) of brace (prescribed wearing time(s) and brace type(s)) used
**4**	Outcome	A definition of success rate (all definitions of success rate were accepted in the qualitative synthesis in this review)

**Table 3 jcm-10-02145-t003:** The MINOR tool [15]. Items 1–8 are for both comparative and non-comparative studies. Items 9–12 are only for comparative studies. The items are scored 0 (not reported), 1 (reported but inadequate), or 2 (reported and adequate), the global ideal score being 16 for non-comparative studies and 24 for comparative studies. For comparative studies: <12 high risk of bias, 12–16 medium risk of bias, >16 low risk of bias. For non-comparative studies: <7 high risk of bias, 7–11 medium risk of bias, >11 low risk of bias.

1	A clearly stated aim	The question address should be precise and relevant.
2	Inclusions of consecutive patients	All patients potentially fit for inclusion had been included in the study.
3	Prospective collection data	Data were collected according to a protocol established before the beginning of the study.
4	Endpoints appropriate to the aim of the study	Unambiguous explanation of the criteria used to evaluate the main outcome.
5	Unbiased assessment of the study endpoint	Blind evaluation of objective end-points and double blind-evaluation of subjective endpoints. Other explanation of the reasons for not blinding.
6	Follow-up period appropriate to the aim of the study	The follow-up should be should be sufficiently long to allow the assessment of the main end-points.
7	Loss to follow-up less than 5%	All patients should be included in the follow-up. Otherwise, the proportion lost should not exceed the proportion experiencing the major end-points.
8	Prospective calculation of the study size	Information of the size of detectable difference of interest with a calculation of 95% confidence interval.
9	An adequate control group	Having a gold standard diagnostic test or therapeutic intervention recognized as the optimal intervention according to the available published data.
10	Contemporary groups	Control and studied group should be managed during the same time period.
11	Baseline equivalence of groups	The groups should be similar regarding criteria and studied end-point.
12	Adequate statistical analysis	Whether the statistics were in accordance with the type of study with calculation of confidence interval or relative risk.

**Table 4 jcm-10-02145-t004:** Overview of the different braces included in this systematic review.

Brace Type	Rigidity	Prescribed Wearing Time
Boston [22,23,24]	Rigid brace	Full-time/part-time
Cheneau brace [25,26,27,28]	Rigid brace	Full-time/part-time
PASB (Progressive Action Short Brace) [29]	Rigid brace	Full-time
Lyon brace [30]	Rigid brace	Full-time
Gensingen Brace [31]	Rigid brace	Full-time
OMC (Osaka Medical College) brace [32]	Rigid brace	Full-time
Pressure-adjustable orthosis [33]	Rigid brace	Full-time
Charleston brace [34,35]	Rigid brace	Night-time
Providence brace [5,36,37,38,39]	Rigid brace	Night-time
SpineCor [40,41]	Soft brace	Full-time

**Table 5 jcm-10-02145-t005:** Overview of the non-comparative studies. The studies lightened in bold are the ones included in the meta-analysis.

First Author	Year	Risk of Bias	Sample Size	Cobb Angle	Skeletal Maturity	Type of Brace	Brace	Timing	Follow-Up	Definition of Success Rate	Success Rate
Weinstein [6]	2014	14/24	146	20–40	Risser 0–2	Rigid	TLSO	Full-time	7 years	>50°	72%
			96			Control group					42%
**Xu** [22]	**2019**	**8/16**	**90**	**40–45**	**Risser 0–3 (divided in subgroups)**	**Rigid**	**Boston brace**	**Full-time**	**2 years**	**≤5°**	**51.1%**
Yrjonen [23]	2007	10/24	51	>25°	Risser 0–3	Rigid	Boston brace	Full-time	> 1 year	≤5°	Girls 78.4%
			51								Boys 62.7%
**Grivas** [24]	**2003**	**7/16**	**28**	**20–40**	**Pre or < 1 year post-menarche and Risser**	**Rigid**	**modified Boston brace**	**Full-time**	**mean of 2.3 years**	**≤5°**	**82%**
Pasquini [25]	2016	5/16	843	20–40	Risser 0–2	Rigid	modified Cheneau brace	Full-time	≥2 years	≤5°	81%
Fang [26]	2015	10/16	32	25–40	Risser 0–2	Rigid	Cheneau brace	Full-time	2 years	no curve progression ≥ 50°	81%
Pham [27]	2007	7/16	63	20–45	Risser 0–2	Rigid	Cheneau brace	Full-time	2 years after discontinuing brace therapy	<10°	85.7%
**Zabrowska Sapeta** [28]	**2010**	**7/16**	**79**	**20–45**	**Risser 0–4**	**Rigid**	**Cheneau brace + exercises**	**Full-time**	**1–5 years**	**≤5°**	**48%**
**Maruyama** [44]	**2015**	**9/16**	**33**	**25–40**	**Risser 0–2 and pre or 1 year post-menarche**	**Rigid**	**Rigo- Cheneau brace**	**Full-time**	**Mean 2.8 years**	**≤5°**	**76%**
Aulisa [29]	2009	6/16	50	25–40	Risser 0–2	Rigid	PASB	Full-time	> 2 years	≤5°	100%
**Aulisa** [45]	**2020**	**12/16**	**163**	**20–60** **mean 28**	**Risser 0–4**	**Rigid**	**PASB**		**10y after termination**	**≤5°**	**65.6%**
Aulisa [30]	2015	4/16	69	25–40	Risser 0–2	Rigid	Lyon brace	Full-time	2 years	≤5°	98.5%
**Weiss** [31]	**2017**	**9/16**	**25**	**≥40**	**Risser 0–2**	**Rigid**	**Gensingen Brace**	**Full-time**	**≥ 1.5 years**	**≤5°**	**92%**
Kuroki [32]	2015	10/16	31	20–40	Risser 0–2	Rigid	OMC brace	Full-time	2 years after discontinuing brace therapy	no curve progression ≥ 50°	67.8%
Yangmin Lin [33]	2020	6/16	24	20–40	Risser 0–2	Rigid	Pressure-adjustable orthosis		1 year	≤5°	100%
**Lateur [46]**	**2017**	**10/16**	**142**	**<25**	**Risser 0–3**	**Rigid**	**Night-time brace**	**Night-time**	**>1 year mean 3.75 y**	**≤5°**	**83%**
**Price [34]**	**1990**	**7/16**	**139**	**25–49**	**Risser 0–2**	**Rigid**	**Charleston brace**	**Night-time**	**> 1 year**	**≤5°**	**83%**
**Lee [35]**	**2012**	**9/16**	**95**	**25–40**	**Risser 0–2**	**Rigid**	**Charleston brace**	**Night-time**	**> 2 years after skeletal maturity**	**≤5°**	**84.2%**
Davis [36]	2019	6/16	56	25–40	Risser 0–2	Rigid	Providence brace	Night-time	mean 2.21 years	≤5°	51.8%
**Ohrt-Nissen [37]**	**2016**	**7/16**	**63**	**25–40**	**Risser 0–2**	**Rigid**	**Providence brace**	**Night-time**	**2 years**	**≤5°**	**57%**
**D’ Amato [5]**	**2001**	**8/16**	**102**	**20–42**	**Risser 0–2**	**Rigid**	**Providence brace**	**Night-time**	**Min 2 y after stop bracing**	**≤5°**	**74%**
Bohl [38]	2014	6/16	34	25–40	Risser 0–2	Rigid	Providence brace	Night-time	2 years after maturity	≤5° or >45 degrees	50% >5°, 59% >45°
**Simony** [39]	**2019**	**10/16**	**80**	**20–45**	**Pre or < 1 year post-menarche**	**Rigid**	**Providence brace**	**Night-time**	**Till 1 year after stop bracing**	**≤5°**	**89%**
**Coillard** [40]	**2007**	**8/16**	**170**	**25–40**	**Risser 0–2**	**Soft**	**SpineCor**	**Full-time**	**2 years after discontinuing brace therapy**	**≤5°**	**59.4%**
**Coillard** [41]	**2014**	**16/24**	**32**	**15–30**	**Risser 0–2**	**Soft**	**SpineCor brace**	**Full-time**	**5 years**	**≤5°**	**73%**
			**36**				**control group**				**57%**

**Table 6 jcm-10-02145-t006:** Overview of the comparative studies between braces. The studies lightened in bold are the ones included in the meta-analysis.

First Author	Year	Risk of Bias	Sample Size	Cobb Angle	Skeletal Maturity	Type of Brace	Brace	Timing	Follow-Up	Definition of Success Rate	Success Rate
Minsk [47]	2017	11/24	13	25–40	Risser 0–2	Rigid	Rigo- Cheneau	Full-time	>1 year	≤5°; no need of surgery	Spinal surgery: 0% >6°:31%
			93				Boston				Spinal surgery: 34%
Hanks [42]	1988	11/24	75	>25	Risser 0–4	Rigid	Wilmington Jacket	Full-time	1 year after discontinuing brace	<10°	Full-time 80%
			25					Part-time			84%
Katz [43]	2010	11/24	57	25–40	Risser 0–2	Rigid	Boston brace	Full-time	>1 year	≤5°	82% > 12 h
			43					Part-time			31% > 7 h
**Yrjonen** [4]	**2006**	**12/24**	**36**	**>25°**	**Risser sign 0–3**	**Rigid**	**Providence brace**	**Night-time**	**mean 1.8 years**	**≤5°**	**72%**
			**36**			**Rigid**	**Boston brace**	**Full-time**			**78%**
Janicki [48]	2007	10/24	35	25–40	Risser 0–2	Rigid	Providence brace	Night-time	>2 years	≤5°	31%
			48			Rigid	Custom TLSO	Full-time			15%
Ohrt-Nissen [49]	2019	13/24	40	25–40	Risser 0–2	Rigid	Providence brace	Night-time	2 years	≤5° (primary outcome); curve progression ≥45°	45%
			37			Rigid	Boston brace	Full-time			38%
Weiss [50]	2005	8/24	12	15-30 and >30 for rigid brace	Risser sign 0 (one exeption with 1) Tanner 2 or 3	Soft	SpineCor	Full-time	mean 3.5 years	≤5°	8%
			10			Rigid	Cheneau brace	Full-time			80%
**Guo** [51]	**2014**	**13/24**	**20**	**20–30**	**Risser 0–2 Pre or < 1 year post-menarche**	**Soft**	**SpineCor brace**	**Full-time**	**2 years after discontinuing brace therapy**	**≤5°**	**65%**
			**18**			**Rigid**	**TLSO**	**Full-time**			**94%**

## Data Availability

Data is contained within the article or Appendix A. The data presented in this study are available in “Appendix A The Effectiveness of Different Concepts of Bracing in Adolescent Idiopathic Scoliosis (AIS) A Systematic Review and Meta-Analysis”.

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
