# Peer review of "The Effectiveness of Different Concepts of Bracing in Adolescent Idiopathic Scoliosis (AIS): A Systematic Review and Meta-Analysis"

_jcm, 2021, doi:10.3390/jcm10102145_

Round 1

Reviewer 1 Report

Concerns have been adressed sufficiently.  Recommend for publication 

Reviewer 2 Report

The Authors satisfactorily addressed all my comments, therefore I'm glad to support the revised paper for Publication.

This manuscript is a resubmission of an earlier submission. The following is a list of the peer review reports and author responses from that submission.

Round 1

Reviewer 1 Report

Dear Authors: This is a very useful  concept to develop . I have some concerns that many of your 16 articles did have documented brace wear time using a data logger, such as iButton.

The Weinstein article reported success based on reaching a surgical level of 50 degrees. However, the other studies used >5 degrees . So data from Weinstein should be be used. 

Reviewer 2 Report

Well designed and performed meta-analysis of bracing in AIS. All (in my opinion) aspects of bracing were analysed in a appropriate way. The authors should stress what their meta-analysis added to the current knowlege.

Much less comprehensive meta-analysis was published in 2019 (Zhang Y, Li X. Treatment of bracing for adolescent idiopathic scoliosis patients: a meta-analysis. Eur Spine J. 2019;28(9):2012-2019.

Reviewer 3 Report

The study is a systematic review on effectiveness of different bracing concepts in conservative treatment of AIS. The study is well written and has a clear and systematic methodology. 

Specific comments

Line 41: "life-long burden". I suggest rephrasing as burden indicates severe morbidity, which is not the case for most AIS patients. 

Line 48-50: This is an inappropriate reference to a very small study that is not designed to assess "ideal" og "expected" in-brace correction. I would suggest adding larger studies as reference e.g. D'Amato 2001 for Providence in-brace correction. 

Line 68-70: This is a controversial statement and reference to a dutch study that is not available in english is not sufficient. I suggest adding more/other references, i.e. Abraham NS, Byrne CJ, Young JM, Solomon MJ. Meta-analysis of well-designed nonrandomized comparative studies of surgical procedures is as good as randomized controlled trials. J Clin Epidemiol. 2010;63:238-45.

Line 88:  "Longitudinal studies with at least one-year follow-up". One year from what, brace initiation or termination?

Line 346: "Risser sign is still the most used parameter for bone maturity and initiating treatment in phases 0-2 and 0-3 appeared most effective." I don't understand this conclusion. Compared to what? Risser 4. The analysis carried out in this study does not merit conclusions as to the appropriate Risser grade for bracing. Yes, success rate is higher for Risser 3 but that is because the risk of progression is lower. Or maybe I misunderstanding the conclusion? 

Reviewer 4 Report

I enjoyed reading this Manuscript. Few points, requiring some minor clarification, are provided below to help improving the overall clarity of the paper.

Abstract:

  • line 26 "MINORS": Please, briefly clarify the scope of this tool, as the Reader may not be totally familiar with it.

Introduction:

  • line 44 "brace concepts and specific brace types": I find that talking about "concepts" and "types" may be somehow redundant/misleading, as they seem to address the same aspect. If this is not the case, please, clarify. I personally found the explanation provided in the Abstract (lines 26-27 "Brace concepts... of the brace") quite effective and clear.
  • lines 62-63"concept" vs. "type": Please, have a look to the previous comment.
  • line 66 "RCTs": Although it should be already very clear, please, spell the acronym.
  • lines 6870 "Meta-...discarded": Not clear whether you are referring to any specific study. Please, add a reference to this sentence.
  • line 74 "these types of study": Which ones?

Materials and Methods:

  • line 99 "two authors": Is it enough?

Results:

  • lines 154-157 "Night-time braces", "Soft full-time", "soft", "rigid full-time": Am I right to assume that when only the wearing time or the rigidity of the brace is provided, no other information is available in these study?
  • Figure 3: Clarity is somehow lacking in these graphs. Please, add the title on each axis. If the core message is about highlighting the brace concept (rigidity, wearing time), then I would recommend to report it as a legend, just before each group of Authors et al. YEAR. Please, distinguish the two papers from Guo et al.
  • Figure 4: Please, report the title on the vertical axis.

Discussion:

  • line 319 "beacause of this trial 5": Which trial? Is the little "5" there referring to Reference 5? Please, correct.
  • lines 338-340: This sentence sounds quite redundant ("combination of curve type, initial curve severity and skeletal maturity" vs. "initial Cobb angle, type of curve, sex and skeletal maturity"): Please, amend it.